# Learning for Data Synthesis: Joint Local Salient Projection and Adversarial Network Optimization for Vehicle Re-Identification

**DOI:** 10.3390/s22239539

**Published:** 2022-12-06

**Authors:** Yanbing Chen, Wei Ke, Wei Zhang, Cui Wang, Hao Sheng, Zhang Xiong

**Affiliations:** 1Faculty of Applied Sciences, Macao Polytechnic University, Macao 999078, China; 2School of Computer Science and Technology, Zhejiang Sci-Tech University, Hangzhou 310018, China; 3State Key Laboratory of Virtual Reality Technology and Systems, School of Computer Science and Engineering, Beihang University, Beijing 100191, China; 4Beihang Hangzhou Innovation Institute Yuhang, Hangzhou 310023, China

**Keywords:** vehicle re-identification, sample synthesis, adversarial module, parameter agent network, convolutional neural network, self-attention mechanism

## Abstract

The problem of vehicle re-identification in surveillance scenarios has grown in popularity as a research topic. Deep learning has been successfully applied in re-identification tasks in the last few years due to its superior performance. However, deep learning approaches require a large volume of training data, and it is particularly crucial in vehicle re-identification tasks to have a sufficient amount of varying image samples for each vehicle. To collect and construct such a large and diverse dataset from natural environments is labor intensive. We offer a novel image sample synthesis framework to automatically generate new variants of training data by augmentation. First, we use an attention module to locate a local salient projection region in an image sample. Then, a lightweight convolutional neural network, the parameter agent network, is responsible for generating further image transformation states. Finally, an adversarial module is employed to ensure that the images in the dataset are distorted, while retaining their structural identities. This adversarial module helps to generate more appropriate and difficult training samples for vehicle re-identification. Moreover, we select the most difficult sample and update the parameter agent network accordingly to improve the performance. Our method draws on the adversarial networks strategy and the self-attention mechanism, which can dynamically decide the region selection and transformation degree of the synthesis images. Extensive experiments on the VeRi-776, VehicleID, and VERI-Wild datasets achieve good performance. Specifically, our method outperforms the state-of-the-art in MAP accuracy on VeRi-776 by 2.15%. Moreover, on VERI-Wil, a significant improvement of 7.15% is achieved.

## 1. Introduction

Vehicle re-identification (ReID) [1], including cross-camera recognition and vehicle placement and tracking, is critical for road safety and intelligent transportation [2]. Re-identification refers to recognizing the same vehicles in non-overlapping view domains and across camera networks [3]. It is also defined as the process of identifying a destination vehicle from images or videos captured by video angles without using the license plate [4]. Deep learning has become popular over the last decade, and the best performing vehicle ReID algorithms based on appearance information are now mostly created using these methods [5]. However, deep learning networks need a large amount of training data. The lack of data is an obvious stumbling block, and it leads to overfitting and poor generalization. Unfortunately, data annotation and collection are quite expensive. One effective technique to obtain more training images is data augmentation. In fact, data augmentation has always been an important method in vehicle ReID due to the emergence of deep learning [6]. Regular methods create additional augmented samples through data warping, such as geometric and color transformations. In contrast to the previous traditional techniques [7], our novel method increases the complexity rather than the size of dataset. This framework not only combines geometric augmentation and deep learning technology but also utilizes an attention mechanism. One of the most important basic functions of a framework is to locate the salient region of a sample. However, many existing methods, e.g., Zhou et al. [8], Grad-CAM [9], and Grad-CAM++ [10], focus on the classification tasks. In terms of vehicle ReID, the commonly used methods of RAM [11,12] are only applied to certain model layers. As a result, this paper proposes a unique method for locating the salient region for vehicle ReID, dubbed self-attention. To select a salient region of the image, the region must first be identified, which is crucial for the feature extraction phase. To obtain salient maps, the Spatial Relation-Aware Attention [13] uses the gradients of the confidence score. In turn, the salient region of the image is selected to learn more. First is a geometric augmentation. A neural network is employed to learn the weights for transformation. Next, the transformation states are generated from these weights. To make the network learning more challenging, an adversarial module is created, which consists of the recognizer and parameter agent network mentioned above. This module maintains a balance between the complexity and identity of a vehicle image. Figure 1 depicts the changes from an original to an augmented image.

As a validation, an ablation test was designed on the VeRi-776 [14], VehicleID [15], and VERI-Wild [16] datasets. Our contribution can be summarized by the following key points.

Incorporation of an attention mechanism to locate the perspective region and the intensity of the perspective.Data augmentation by increasing the difficulty rather than adding more images, without losing the original structure of a dataset.An innovative framework that combines an attention mechanism, geometric data augmentation, and deep learning.An innovative adversarial strategy that integrates the salient projection region location and the local region projection transformation.

The paper proceeds as follows. Section 2 discusses the related work. The research is presented detail in Section 3. Section 4 presents the experiments and discussion. In Section 5, we conclude our work and provide an outlook on future work.

## 2. Related Work

### 2.1. Vehicle ReID

Vehicle ReID recognizes a target vehicle across multiple cameras with non-overlapping views. Generally speaking, ReID requires robust and discriminative representations [17]. In the past few years, Liu et al. [14] researched large-scale bounding boxes. They merged color, texture, and deep learning to elevate the semantic relations. Li et al. [18] proposed a DJDL model to extract discriminative representations. Shen et al. [19] suggested a two-step structure to effectively incorporate complex spatial–temporal vehicle data for generalization. The MGN [20] model extracted global and local features from an object using a multi-branch network from a multiscale perspective. Stevenson et al. [21], on the other hand, proposed an effective part-regularized discriminative model. This method expanded the ability to sense differences to good effect. GSANet is based on the idea of SCAN [22], and it solved the problem of information imbalance by using channel attention mechanisms and spatial attention mechanisms.

### 2.2. Generative Adversarial Networks

In recent years, the most advanced image generators [23,24,25,26] have been the generative adversarial networks (GAN) [27]. GAN has achieved success in image generation, relying on the continuous improvement of the GAN modeling ability under gaming to finally achieve fake-like image generation. It is a contest in which a generator and a discriminator compete. The generator creates new samples, and the discriminator determines determine whether the synthesized sample should be retained or discarded. GAN is a powerful data augmentation technology in fields with a lack of image samples. It can help to generate artificial scenarios while preserving features similar to the original dataset. GAN has been described by Bowles et al. [28] as a method for “unlocking” additional information. DCGAN [29], CycleGAN [30], and Conditional GAN are a few of improvements proposed to expand the GAN concept. The quality of the samples generated has been significantly improved as a result of these advancements.

### 2.3. Data Augmentation

Data augmentation [6] in deep neural network training is conducted to avoid overfitting [31]. As we know, differing views, illumination, low contrast, background, and scales present a challenge for vehicle ReID. Nevertheless, there are not many good solutions to address the challenges. The aim of data augmentation is to incorporate these translational inconsistencies. It is clear that larger datasets result in better performance [32]. Manually collecting and tagging the data, however, is a difficult process. Many surveys evaluating the effectiveness of data augmentation employ common academic image datasets as the benchmark. Many augmentation techniques, such as swapping, inversion, resizing, and perspective transformation [6], are strictly outlined as basic methods. Our proposed method is to mix those methods with neural networks to machine-learn instead of manually adjusting the parameters.

### 2.4. Salient Region Locating

A salient region locating method should indicate which pixels of the image are more vital. It can significantly support the evaluation of the model and optimization of its performance. Methods including CAM [8], GCAM [9], and GCAM++ [10] are intended for classification tasks. GCAM and GCAM++ generate a location outline that emphasizes the important areas in the image, while being influenced by transmission line losses using the gradients of the target label. There are two main methods for locating salient regions in vehicle ReID tasks. The first is RAM [11], which requires a GAP between the very last prediction model and the set of inputs bounding the salient areas. Another method [12] converts a recall task into a classification task using the siamese network, so it can only be employed in the siamese network [33].

## 3. Methodology

An adversarial strategy is proposed to use projection transformation to generate samples for vehicle ReID. Section 3.1 explains the overall structure. Then, we present the salient projection region location (Section 3.2) and the local region projection transformation (Section 3.3). Finally, Section 3.4 describes the transformation state adversarial module.

### 3.1. Overall Framework

As shown in Figure 2, our model has three parts: salient projection region location (SPRL), local region projection transformation (LRPT), and transformation state adversarial module (TSAM). First of all, we enter an original vehicle image, denoted as *P*. Then, the SPRL takes the input image *P* to locate the salient region. Next, the LRPT produces 11 transformed images $x_{1},\cdots ,x_{11}$. Finally, given the transformed images, TSAM selects the most difficult sample among them from the input image *P*. The recognizer and the parameter agent network (PAN) are the two major components of the TSAM. When the augmented images $x_{1},\cdots ,x_{11}$ pass through the recognizer, each augmented image $x_{i}$ will be judged on whether it retains the same identity as *P*. The generated images will be discarded if they belong to a different identity. Note that PAN is present in both the LRPT and TSAM modules. It is utilized in the LRPT to create the weights needed for the projection transformation. The PAN, as part of the TSAM, is intended to create the most difficult generated sample. Finally, the most difficult sample selected by TSAM will replace the input image.

### 3.2. Salient Projection Region Location

The salient projection region location (SPRL) is used to extract a region of interest. It projects the local region of an image for further processing. SPRL combines the random rectangle selection and Spatial Relation-Aware Attention [13] to locate the local region of the input image. The obtained local region is then fed into the LRPT to transform the image. We want to focus attention on those image regions with relatively larger weights; thus, after the perspective transformation on the subsequent focused regions, the noise of the image is significantly increased. By increasing the complexity of the training images, the robustness of the network improves. Naturally, because the Spatial Relation-Aware Attention Network can obtain the salient region, it is regarded as a method in the test stage. Figure 3, Column 2, depicts some of the visualization results.

Furthermore, because vehicle ReID is a zero-shot learning problem [34], which means that recognition in the analysis stage does not take place in the training stage, many existing methods [8,9,10] intended for classification that need identities to find the salient regions cannot be applied in the test stage. However, our method does not have this issue. The algorithm of the SPRL is shown in Algorithm  1. In our study, we assume the area ratio as $A_{t}$ and the aspect ratio as $R_{t}$; then, we obtain the width $W_{t}$ and height $H_{t}$ from $A_{t}$ and $R_{t}$. The salient region is partly determined by the top-left corner *P*. In addition, the salient region is obtained by $\left(P,W_{t},H_{t}\right)$.
**Algorithm 1:** Salient Projection Region Location Procedure**Input:** image *P*, area *A*, ratio of width and height *R*, area ratio ranging from $A_{1}$ to $A_{2}$,  aspect ratio ranging from $R_{1}$ to $R_{2}$;**Output:** selected region 1: W←P.width 2: H←P.height 3: $A_{t}\leftarrow \mathrm{rand}\left(A_{1},A_{2}\right)\times A$ 4: $R_{t}\leftarrow \mathrm{rand}\left(R_{1},R_{2}\right)$ 5: $H_{t}\leftarrow \sqrt{A_{t}\times R_{t}}$ 6: $W_{t}\leftarrow \sqrt{A_{t}÷R_{t}}$ 7: Weight←SRA(P) 8: SRA is Spatial Relation-Aware Attention [13]. The weight matrix *W* with ${\mathrm{Weight}}_{P}$ rows  and ${\mathrm{Height}}_{P}$ columns can be obtained from the SRA [13]. Obtain the weights. 9: **for** *i* In $W_{t}$ **do**10:    **for** *j* In $H_{t}$ **do**11:      12:      **if** $i+W_{t}\leq W\wedge j+H_{t}\leq H$ **then**13:         $S\leftarrow \sum_{i}^{i+W_{t}}\sum_{j}^{j+H_{t}}\mathrm{Weight}$14:      **end if** 15:    **end for**16: **end for**17: Use the loop to find the maxS and its P(x,y)18: return region $\left(P,W_{t},H_{t}\right)$

### 3.3. Local Region Projection Transformation

The local region projection transformation (LRPT) produces 11 augmented images. This module involves two major processes: the parameter agent network (PAN) and the projection transformation. The transformation parameters are generated by the PAN. Combining the salient rectangle region and the parameters mentioned above, we then employ the projection transformation process to generate the augmented images.

#### 3.3.1. Parameter Agent Network

The image is supplied to PAN once the local area has been traced out from the original image. PAN constructs the parameters for the projection transformation.

Table 1 shows that PAN has six convolutional and one fully connected layer. MP denotes the maximum pooling, while BN represents the batch normalization. Finally, the FC layer generates 10×2 parameters. The 10 pairs of parameters are shown in the red part of Figure 4, and each pair is written as $\left(x^{\prime },y^{\prime }\right)$. The remaining points are obtained from the rectanglular interest region of the image using the 10 pairs of parameters, see Section 3.3.2 for more details. A transformation state is made up of the 10 pairs of parameters and their weights, and the augmented images can be generated with these transformation states.

In order to generate the 11 augmented samples, 10 other transformation states are obtained ranging from $s_{2}$ to $s_{11}$. Other states are generated from state $s_{1}$ by choosing one point in $s_{1}$ and switching its coordinate in state $s_{1}$ to the opposite direction. Specifically, as shown in Figure 5, each of the 10 points in state $s_{1}$ is selected. Then, we switch it to the opposite direction by using the nearest vertical edge of the rectangle as the axis. Thus, the direction of this point is flipped to update the *x*-coordinate of the point, while the *y*-coordinate remains unchanged. Since $s_{1}$ has 10 coordinates, we can generate in total 10 other states, $s_{2},\cdots ,s_{11}$, by this method. Finally, we apply the projection transformation process to the 11 states, $s_{1},\cdots ,s_{11}$, and combine the original image *P* and the states to generate 11 augmented images, $x_{1},\cdots ,x_{11}$, as the result of the LRPT.

#### 3.3.2. Projection Transformation

After achieving the 11 transformation states, the projection transformation can be used to convert the input image *P* into the augmented samples. As shown in Figure 4, the projection transformation will project the image onto a new coordinate screen. The original points are transformed to other points on the projected screen, and the image is transferred as a result of the projection transformation, but the image pixel information is preserved. The details are as follows. All pixels should be relocated in accordance with the projection mapping rule shown below once the anchor points have been established. To generate an augmented image, it applies similarity deformation based on moving the least squares [35] in the input image. Suppose the pixel to be moved is u=(x,y), and the target moved position is $t=(x^{\prime },y^{\prime })$. *t* can be obtained from *u* by
(1)$$t\leftarrow \left(u-p^{*}\right)\times M+q^{*}.$$

In Equation (Equation 1), $M\in R^{2\times 2}$ is a matrix [36] having the property $M^{T}\times M=\lambda^{2}I$, where λ takes values within a range depending on the specific situation. $p^{*}$ and $q^{*}$ are the weights developed from the 10 anchor points $p_{i}=\left(x_{i},y_{i}\right)$ from the input image and the 10 transformed anchor points $q_{i}=\left(x_{i}^{\prime },y_{i}^{\prime }\right)$ in the transferred image, respectively. To summarize, the values of *i* in Equation (Equation 1) are within 1,⋯,10. Our algorithm also needs to obtain the weighted values corresponding to these points,
(2)W←SRA(P),
where SRA stands for the Spatial Relation-Aware Attention [13], which consists of the values representing the weights. The weight matrix *W* with ${\mathrm{Weight}}_{P}$ rows and ${\mathrm{Height}}_{P}$ columns can be obtained from the SRA; then, the specific weighted values are obtained by coordinate correspondence. The variable *P* in (Equation 2) represents the original image. Formally, we have
(3)$$p^{*}\leftarrow \left(\frac{\sum x_{i}W_{x_{i},y_{i}}}{\sum W_{x_{i},y_{i}}},\frac{\sum y_{i}W_{x_{i},y_{i}}}{\sum W_{x_{i},y_{i}}}\right),$$
(4)$$q^{*}\leftarrow \left(\frac{\sum x_{i}^{\prime }W_{x_{i},y_{i}}}{\sum W_{x_{i},y_{i}}},\frac{\sum y_{i}^{\prime }W_{x_{i},y_{i}}}{\sum W_{x_{i},y_{i}}}\right).$$

As shown in Table A1, the inclusion of a table defining the variables enables a more specific understanding of the variables. The augmented image can be obtained after the appropriate coordinates of all the pixel points have been generated.

### 3.4. Transformation State Adversarial Module

TSAM transfers a sample that is as near to the original as is practical while maintaining quality. The recognizer, the learning target selection, and the parameter agent are the three components of the TSAM framework. The structure of the TSAM is depicted in the green section of Figure 2. Unlike other methods that use deep learning to find the best policies, this adversarial module uses adversarial processing to generate a properly transformed image. Algorithm 2 describes the entire TSAM process. This module is made up of the PAN and the recognizer. The PAN produces the transformation states, which are then used to generate the augmented images. The recognizer confirms that the augmented image label has not lost its identity and chooses the most difficult image.
**Algorithm 2:** Adversarial Process of Parameter Agent Network (PAN) and Recognizer**Input:** input image *P*, selected region *A***Output:** optimized ${\mathrm{PAN}}^{\prime }$ 1: sample transformation state $s_{1}$ from predicted distribution: $s_{1}\leftarrow \mathrm{PAN}\left(A\right)$ 2: generate random transformation states from $s_{2}$ to state $s_{11}$, based on state $s_{1}$ 3: all states including $s_{1}-s_{11}$ contain anchor points 4: obtain augmented images from $x_{1}$ and $x_{11}$ based on states $s_{1}$ to state $s_{11}$: 5: The following i all represent 1 to 11 6: $x_{i}\leftarrow \mathrm{PT}\left(P,s_{i}\right)$ 7: identify images: Reg←Recognizer(P) ${\mathrm{Reg}}_{i}\leftarrow \mathrm{Recognizer}\left(x_{i}\right)$ 8: **if** 
$\mathrm{ID}\left[{\mathrm{Reg}}_{i}\right]=\mathrm{ID}\left[\mathrm{Reg}\right]$ **then** 9:    test distance Δ between ${\mathrm{Reg}}_{i}$ and Reg10:    **if** $\max\Delta ({\mathrm{Reg}}_{i},\mathrm{Reg})$ **then**11:      ${\mathrm{PAN}}^{\prime }$ = update PAN with $s_{i}$12:    **end if**13: **else**14:   ${\mathrm{PAN}}^{\prime }$ = PAN15:
**end if**


#### 3.4.1. Recognizer

The recognizer verifies that the generated and input samples have the same identity. The recognizer basis network framework is ResNet50 [37]. It is a classification network that has been proven to be extremely efficient. It is used as a recognizer for our network with some adaptation. ImageNet [38] is a database with over 20 thousand different categories. After pretraining on ImageNet [39], it is finetuned to form the recognizer. Then, this recognizer is used to determine the input image. Following that, the recognizer indicates which classification each image belongs to. A transformed sample is compared to the original image to determine whether or not they share the same identity. If the image identity differs from the initial, it is removed from the workflow. If not, it is saved for later use.

#### 3.4.2. Learning Target Selection

During this step, the most difficult augmented image is selected to replace the original image. Meanwhile, the selected state updates the PAN parameters by minimizing the corresponding transformation loss. It is assumed that an augmented image differs from the initial as much as possible. The augmented sample, on the other hand, cannot be so dissimilar to the original that it loses its identity. We calculate the distances between $x_{1},\cdots ,x_{11}$ and *P*, as shown in Figure 6.

The most difficult augmented image is chosen. If only one augmented sample passes, this one is directly adopted. An augmented image with the longest difference is selected and optimized by PAN using the corresponding transformation state with this strategy. Finally, the most difficult image $x^{\prime }$ is chosen and will replace *P* in the dataset.

The loss is described as in Equation (Equation 5), and α is a hyperparameter that controls the loss in a flexible manner.
(5)$$Loss\leftarrow func(x^{\prime },\;x,\;\alpha )$$
Using this strategy, the most difficult augmented image is chosen, and the PAN is optimized.

## 4. Experiments and Discussion

Ablation experiments were designed and compared to the state-of-the-art vehicle ReID models.

### 4.1. Datasets

The datasets for the experiment included the following three well-known datasets.

#### 4.1.1. VehicleID

This dataset has 111,585 photographs of 13,113 automobiles, while the training data have 10,178 images of 13,134 different objects. Three separate sets of varying sizes make up the test set (i.e., Small, Medium, and Large).

#### 4.1.2. VeRi-776

Actual surveillance scenarios collected over 50,000 images. One square kilometer was captured over 24 hours. The images were shot in a free-form monitoring scene and marked with aspects such as type, color, and brand. In summary, this dataset has 776 different vehicles.

#### 4.1.3. VERI-Wild

This dataset was collected in the wild. These samples were captured by 174 cameras over the course of one month (30×24h) in unconstrained scenarios. The differentiation of this dataset is reflected in the illumination and weather changes due to the time span, which can be used for research in areas such as target detection.

### 4.2. Implementation

All the images were resized to 320 px × 320 px at the beginning. ResNet50 [40] served as the backbone network, and the optimizer was the SGD [41]. The loss of network was Soft margin triplet loss [42]. To test the efficacy of our framework, the same settings were retained as the baseline [43]. A value was initialized for the learning rate at the beginning. After the first ten epochs, the learning rate steadily dropped to $10^{-3}$, after starting at $10^{-2}$. The learning rate then changed further as the training progressed. There were 120 training epochs. The experiment contained our strategy in the baseline preprocessing as a data augmentation module, as shown in Figure 7. For further information on the baseline [43], see [44,45].

### 4.3. Ablation Study

This framework included three modules: the SPRL, LRPT, and TSAM. All these modules are related, but they can be selectively applied. Our framework can contain only the SPRL and LRPT modules, or it can contain all modules. The results are shown in Table 2. To assess how well each module performed, a series of successive ablation tests was planned. The tests were run on the three datasets using two combinations: SPRL+LRPT and SPRL+LRPT+TSAM. The various modules produced different results as shown in Figure 8.

#### 4.3.1. Our Model vs. Baseline Model

On the VeRi-776 dataset, Rank1 and MAP outperformed by 0.95% and 3.91%, respectively, when only SPRL + LRPT was used. In other words, relative to the baseline, there was a 1.0% and 5.1% improvement in Rank1 and MAP, respectively. On the VehicleID and VERI-Wild datasets, we designed different experiments for the small, medium, and large sets. On the small VehicleID group, the Rank1 and MAP increased by 1.24% and 0.78%. That means the performance was improved by 1.5% and 1.01% above the baseline. Table 2 also shows an improvement of 2.64% and 2.86% on the medium VehicleID group and 0.67% and 0.64% on the large VehicleID group. On the medium VehicleID group compared to the baseline, the improvement was 3.27% and 3.81%, respectively. Then, on the large VehicleID group compared to the baseline, the improvement was 0.85% and 0.87%, respectively. Our system was more advanced for each result. This showed that the SPRL + LRPT and SRPL+ LRPT +TSAM improved the MAP by 3.91%, 4.96% on the VeRi-776 dataset, respectively; while on the small VehicleID group, there were performance gains of 0.78% and 0.83%, respectively. On the medium VehicleID group, the performance improvement was 2.86% and 2.92%, respectively. Finally, on the large VehicleID group, the improvement was 0.46% and 0.54%, respectively. As shown in Table 2, the main component was the LRPT, when combined with the SPRL+TSAM, the performance was significantly improved.

#### 4.3.2. Internal Comparison

When we only used the SPRL + LRPT, Table 2 shows that our work outperformed the baseline. Then, when combined with the TSAM modules, compared to the SPRL + LRPT module only, it improved the MAP by 1.05% on the VeRi-776 dataset and 0.05%, 0.06%, and 0.08% on the VehicleID groups, respectively. The experiments showed that there was also clear improvement on the VERI-Wild datasets. So, the SPRL + LRPT module played a crucial role. The gradual improvement in the modules’ internal performance justified the design of our framework and the need for it.

### 4.4. Comparison with the SOTA

As shown in Table 3, Table 4 and Table 5, our work clearly excelled and achieved the best results. In addition, on VeRi-776, our model outperformed the second best model in MAP by 2.15% and was improved by 22.9% over the original baseline. It performed well, especially on the VehicleID and VERI-Wild datasets. It outperformed the second best model in MAP by 2.7%, 3.85%, and 7.15% on the three different VERI-Wild sets. The experimental results showed that our method was significantly better than the other methods.

### 4.5. Visualization results

In the previous part of the paper, Figure 9 displayed the Local Region Projection Transformation and Salient Projection Region Location. In this section, the visual results are displayed.

In Figure 10, column 1 is the original input image, and the other columns are the augmented results. In detail, Figure 10a–c shows two vehicle samples from the VeRi-776, VehicleID, and VERI-Wild datasets. The result illustrates that the local part of the image was changed, but the overall features of the image were preserved to the maximum extent due to our framework. When selecting appropriate augmented images, the network performance gradually improved.

## 5. Conclusions

A data synthesis framework was designed mainly for vehicle ReID tasks. An attention mechanism was employed to locate the perspective region and the intensity of the perspective. A projection strategy was used to transform an original image. The image with the greatest feature distance from the candidate images while retaining the identity was chosen. Rather than increasing the size of the dataset, the difficulty of the dataset was increased. Our framework seamlessly integrated with the current state-of-the-art baselines, while keeping the datasets’ structures unchanged.

## Figures and Tables

**Figure 1 sensors-22-09539-f001:**
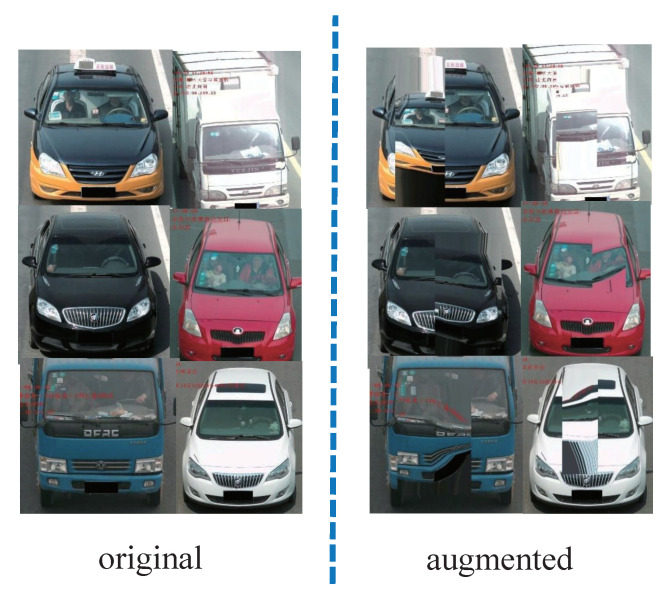
Image augmentation results from an original to an augmented sample.

**Figure 2 sensors-22-09539-f002:**
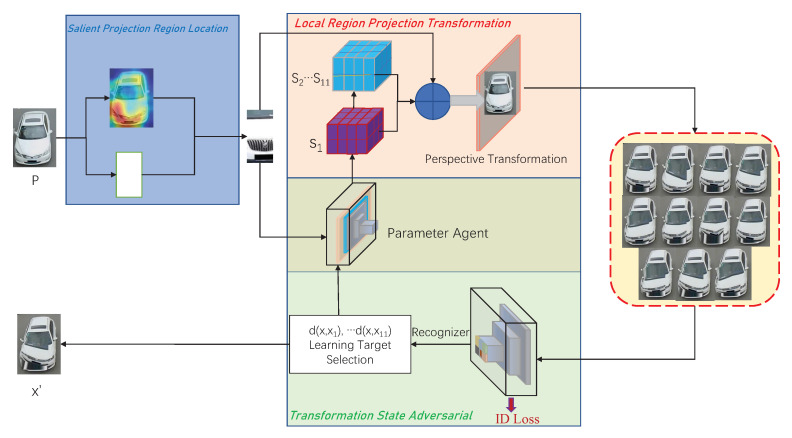
Overall framework. This framework has three parts: Salient Projection Region Location (blue part). Local Region Projection Transformation (red part). Transformation State Adversarial Module (green part).

**Figure 3 sensors-22-09539-f003:**
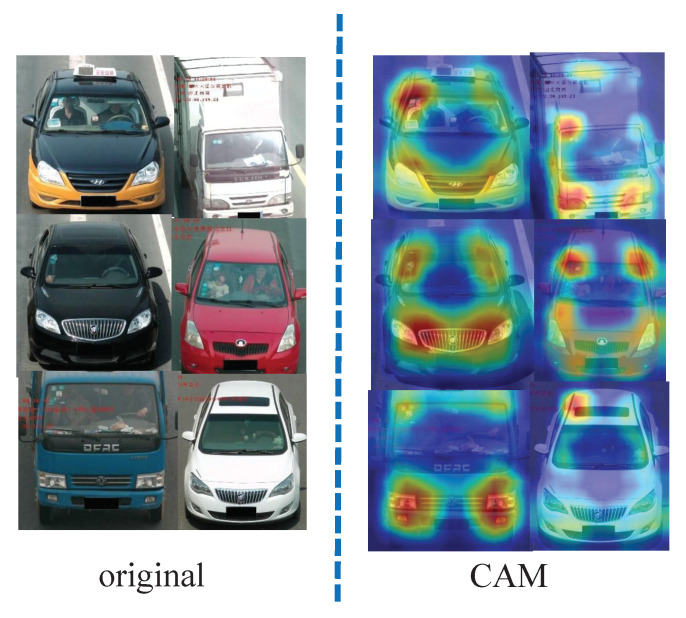
Images results from an original to Spatial Relation-Aware samples.

**Figure 4 sensors-22-09539-f004:**
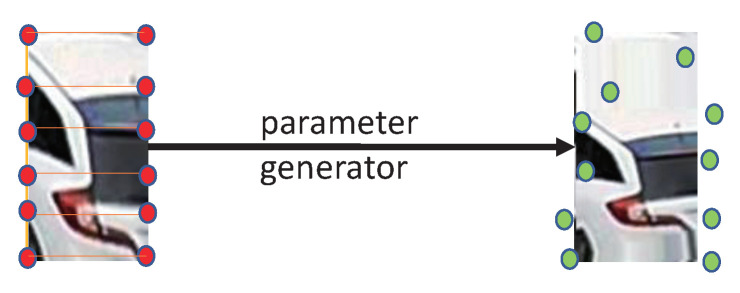
Overview of the Parameter Agent Network.

**Figure 5 sensors-22-09539-f005:**
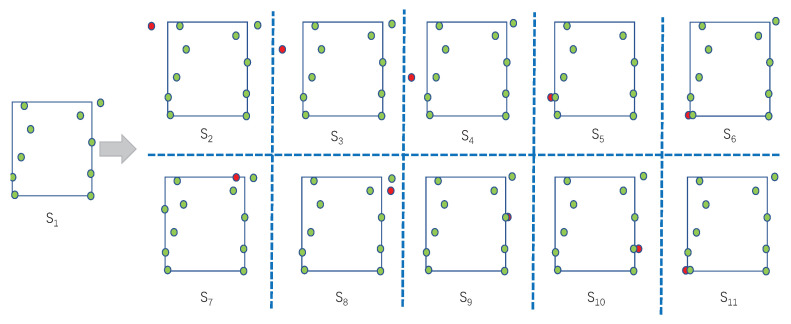
Overview of the transformation state generation process from $s_{1}$ to $s_{2}$,...$s_{11}$.

**Figure 6 sensors-22-09539-f006:**
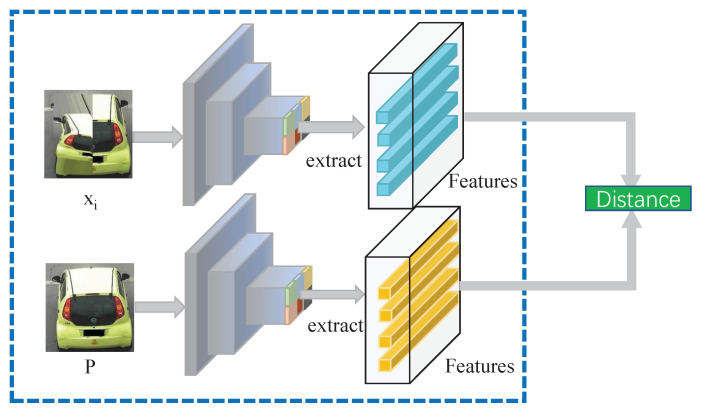
Distance of different images. The augmented samples $x_{i}$ are matched with the input image one by one.

**Figure 7 sensors-22-09539-f007:**
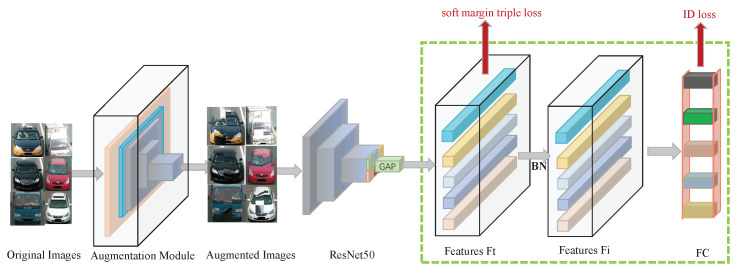
Framework of the Baseline.

**Figure 8 sensors-22-09539-f008:**
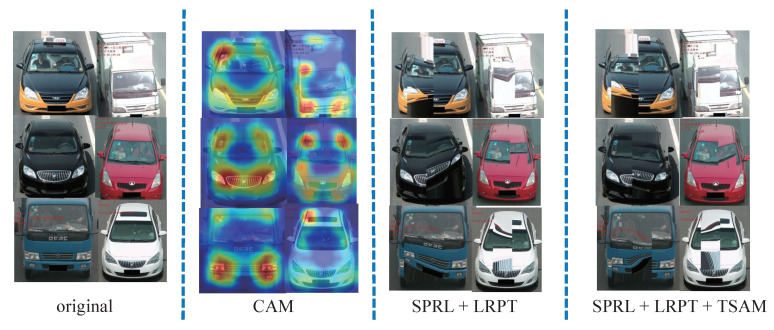
Augmented results with various modules. **Col 1:** Input images. **Col 2:** Heatmap using Relation-Aware Attention. **Col 3:** Augmented results using SPRL + LRPT. **Col 4:** Augmented results using SPRL + LRPT + TSAM.

**Figure 9 sensors-22-09539-f009:**
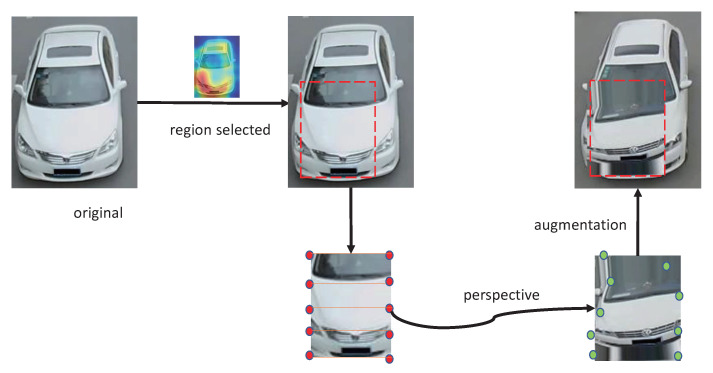
Workflow of the Salient Projection Region Location and Local Region Projection Transformation.

**Figure 10 sensors-22-09539-f010:**
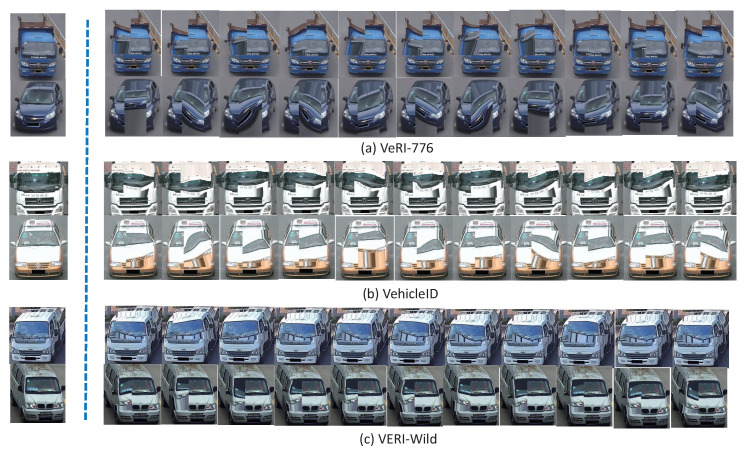
Visualization of the Input Images and the Eleven Augmented Images.

**Table 1 sensors-22-09539-t001:** Architecture of the Parameter Agent Network.

Module	Type
Initial	1×h×w
Conv16, ReLU, MP	16×16×50
Conv64, ReLU, MP	64×8×25
Conv128, BN, ReLU	128×8×25
Conv128, ReLU, MP	128×4×12
Conv64, BN, ReLU	64×4×12
Conv16, BN, ReLU, MP	16×2×6
FC Layer	10×2

**Table 2 sensors-22-09539-t002:** Results from the VeRi-776, VehicleID, and VERI-Wild datasets (%).

Models			VehicleID						VERI-Wild					
	VeRi-776		**Small**		**Medium**		**Large**		**Small**		**Medium**		**Large**	
	**Rank1**	**MAP**	**Rank1**	**MAP**	**Rank1**	**MAP**	**Rank1**	**MAP**	**Rank1**	**MAP**	**Rank1**	**MAP**	**Rank1**	**MAP**
Baseline	95.71	76.59	83.02	77.02	80.74	75.04	79.24	73.98	93.11	72.60	90.54	66.51	86.40	58.52
SPRL + LRPT (Ours)	96.66	80.5	84.26	77.8	83.38	77.9	79.91	74.44	91.81	72.73	90.38	66.64	87.66	58.73
SPRL + LRPT + TSAM (Ours)	96.84	81.55	84.34	77.85	83.42	77.96	79.96	74.52	93.31	72.75	91.03	66.68	87.7	58.78

**Table 3 sensors-22-09539-t003:** Test on VeRi (%).

Models	MAP	Rank1	Rank5
siameseCNN [19]	54.20	79.30	88.90
fdaNet [16]	55.50	84.30	92.40
siameseCNN+ST [19]	58.30	83.50	90.00
provid [46]	53.40	81.60	95.10
aaver [47]	66.35	90.17	94.34
bs [48]	67.55	90.23	96.42
cca [49]	68.05	91.71	94.34
prn [50]	70.20	92.20	97.90
agNet [51]	71.59	95.61	96.56
pamtri [52]	71.80	92.90	97.00
vehicleX [53]	73.26	94.99	97.97
mdl [43]	79.40	90.70	-
ours	**81.55**	**96.84**	**98.99**

**Table 4 sensors-22-09539-t004:** Test on VehicleID (%).

Models	Small		Medium		Large	
	**Rank1**	**Rank5**	**Rank1**	**Rank5**	**Rank1**	**Rank5**
vami [54]	63.10	83.30	52.90	75.10	47.30	70.30
fdaNet [16]	-	-	59.80	77.10	55.50	74.70
agNet [51]	71.15	83.78	69.23	81.41	65.74	78.28
aaver [47]	74.70	93.80	68.60	90.00	63.50	85.60
oife [55]	-	-	-	-	67.00	82.90
cca [49]	75.51	91.14	73.60	86.46	70.08	83.20
prn [50]	78.40	92.3	75.00	88.30	74.20	86.40
bs [48]	78.80	96.17	73.41	92.57	69.33	89.45
vehicleX [53]	79.81	93.17	76.74	90.34	73.88	88.18
ours	**84.34**	**93.48**	**83.42**	**90.63**	**79.96**	**86.52**

**Table 5 sensors-22-09539-t005:** Test on VERI-Wild (%).

Models	Small		Medium		Large	
	**MAP**	**Rank1**	**MAP**	**Rank1**	**MAP**	**Rank1**
googLeNet [56]	24.30	57.20	24.20	53.20	21.50	44.60
fdaNet [16]	35.10	64.00	29.80	57.80	22.80	49.40
mlsl [57]	46.30	86.00	42.40	83.00	36.60	77.50
aaver [47]	62.23	75.80	53.66	68.24	41.68	58.69
bs [48]	70.05	84.17	62.83	78.22	51.63	69.99
ours	**72.75**	**93.31**	**66.68**	**91.03**	**58.78**	**87.70**

## Data Availability

Not applicable.

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
