# Peer review of "Learning for Data Synthesis: Joint Local Salient Projection and Adversarial Network Optimization for Vehicle Re-Identification"

_sensors, 2022, doi:10.3390/s22239539_

Round 1

Reviewer 1 Report

Interesting research into data complexity enhancement permitting deep learning in instances of sparse data, and the work is crafted in a decently drafted manuscript that needs some mild revisions.

·        The manuscript is clear, relevant for the field and presented in a well-structured manner. The cited references are current (mostly within the last 5 years). The manuscript is scientifically sound, and the experimental design is appropriate to test the hypothesis. The manuscript’s results are reproducible based on the details given in the methods section. The figures/tables/images/schemes appropriate and properly show the data. They are easy to interpret and understand. The data is interpreted appropriately and consistently throughout the manuscript. The conclusions are consistent with the evidence and arguments presented.

Abstract is okay but is not likely to entice the readership to continue reading the rest of the manuscript.

·        Use of acronyms/abbreviations in an abstract is unlikely to attract readers not already aware of the manuscript’s content.

·        Results are presented in decent, quantitative fashion, e.g., percent performance improvement compared to a declared benchmark.

Introduction is decently done with some omitted very recent literature and some mild abuse of multi-citation without elaboration.

·        Manuscript neglects the deterministic approaches in favor or narrow review of only stochastic approaches emphasizing deep learning.

·        Please elaborate a reason for the reader to investigate each of the quadruple cited references [23-26] in the introduction.

Equations are scientifically sound and well presented, enhancing the manuscript quality.

Figures and tables are decently done with some mandatory improvements to ensure the readership has access to the content.

·        Internal font size is occasionally too small and/or blurry.  Internal fonts in figures 2,4,7 and table 2 have challenging legibility.

·        Line styles and sizes are identical in figures XXXXX rendering the disparate data indistinguishable when the manuscript is read in printed hardcopy (particularly in black and white) negating the value of the figures due to reliance on colors.

·        Inclusion of a table defining variables and acronyms in an appendix is welcome and effective. Please add such. 

First-person tense should be eliminated.

Reviewer 2 Report

The problem of vehicle re-identification (ReID) in surveillance scenarios is a very trending research topic and the increasing labeled dataset demanding is a problem which has been correctly addressed by the authors with a novel approach, supported by their experiments.
The topic is interesting and fits into the Sensors Journal.
References are new and directly related to the paper. The paper presents a good flow of the research area, but I have minor revisions mostly related to typos, for convenience I have attached the manuscript with these typos highlighted in yellow.
